# Chlorpyrifos Removal in an Artificially Contaminated Soil Using Novel Bacterial Strains and Cyclodextrin. Evaluation of Its Effectiveness by Ecotoxicity Studies

Alba Lara-Moreno [1,2,*] , Esmeralda Morillo [1], Francisco Merchán [2], Fernando Madrid [1] and Jaime Villaverde [1]

1   Department of Agrochemistry, Environmental Microbiology and Soil Conservation, Institute of Natural Resources and Agrobiology of Seville, Spanish National Research Council (IRNAS-CSIC), 41012 Seville, Spain
2   Department of Microbiology and Parasitology, Faculty of Pharmacy, University of Seville, 41012 Seville, Spain
*   Correspondence: alara9@us.es

**Abstract:** The removal of chlorpyrifos (CLP) from the environment is a matter of general interest, because it is one of the most widely used insecticides in the world but presents a high toxicity and persistence in the environment. Biological strategies are considered as a good option to remediate different environmental compartments. Assisted natural attenuation was used to find the ability of different kinds of soils to mineralise CLP. In this way, two soils showed the capacity to degrade CLP (R and LL up to 47.3% and 61.4% after 100 d, respectively). Thus, two CLP-degrading strains, *Bacillus megaterium* CCLP1 and *Bacillus safensis* CCLP2 were isolated from them, showing the capacity to degrade up to 99.1 and 98.9% of CLP in a solution with an initial concentration of 10 mg L$^{-1}$ after 60 d. Different strategies were considered for increasing the effectiveness of soil bioremediation: (i) biostimulation, using a nutrients solution (NS); (ii) bioaugmentation, using *B. megaterium* CCLP1 or *B. safensis* CCLP2; (iii) bioavailability enhancement, using randomly methylated β-cyclodextrin (RAMEB), a biodegradable compound. When bioaugmentation and RAMEB were jointly inoculated and applied, the best biodegradation results were achieved (around 70%). At the end of the biodegradation assay, a toxicity test was used to check the final state of the bioremediated soil, observing that when the degrading strains studied were individually inoculated into the soil, the toxicity was reduced to undetectable levels.

**Keywords:** chlorpyrifos; bioremediation; *Bacillus*; RAMEB; ecotoxicology

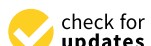



## 1. Introduction

Soil, groundwater, and surface water pollution by pesticides is a global concern. Organophosphorus pesticides have been widely used around the world [1], and these compounds pose a threat to the human health and environment. Among them chlorpyrifos (CLP) [O, O-diethyl O-(3,5,6-trichloro-2-pyridyl) phosphorothioate] is an insecticide, used to control pests in different crops (cotton, fruits, nuts, etc.), lawns, and ornamental plants and it is the fourth most widely applied pesticide in residential, agricultural, and commercial applications due to some characteristics such as its low cost and its high efficiency [2]. Regarding the mechanism of action, it is an acetyl cholinesterase (AChE) inhibitor, an enzyme that hydrolyses the neurotransmitter acetylcholine (ACh) by phosphorylation or phosphonylation of the active site, producing a nervous collapse in the insect [3–6]. However, this insecticide presents also serious risks for humans, since CLP causes neurotoxic disorders, affects the respiratory system and reproductive capacity, and it does not allow the correct development of the brain [7]. Although its use has been banned in many countries, such as those of the UE, it is still used in many South American states, such as Brazil or Mexico, and in Asian countries (China, Bangladesh). In China and the United States of America (USA) it is still used but with restrictions [2]. Its continued application

leads to its accumulation and therefore induces damage to the environment and human health [8]. At the beginning of this century, according to the USEPA (United States Environmental Protection Agency), CLP was located mainly in contaminated water and terrestrial ecosystems [9]. Nowadays, its presence remains in water samples and soils even in those countries where CLP has been banned [10]. Its accumulation could affect soil properties due to the fact that it inhibits nitrogen mineralisation, catalase, and dehydrogenase activity, affecting soil productivity [2].

Microbial degradation is known to be one of the best options for the removal of CLP from the environment [1]. In this process, degrading microorganisms can convert complex organic substances to simpler and smaller structures. This natural process is often hampered by a multitude of parameters, such as the continuous input of pollution, limited availability of nutrients, high concentration of certain contaminants that suppress endogenous microbiota growth, or the original composition of the microbial ecosystem [11]. Natural attenuation is currently an attractive strategy to achieve the bioremediation of contaminated soils due to the action of the endogenous microbiota [12]. This technique is used because of its low cost, however, it is a long process that depends on the kind of contamination and the characteristics and environmental conditions, so that in most cases assisted natural attenuation (ANA) is required by applying different strategies that will help to improve and/or accelerate bioremediation. There is a wide variety of strategies which can help to improve a bioremediation process such as biostimulation [13] or bioaugmentation [14].

CLP biodegradation in aqueous solution has been studied by several authors. Singh et al. [15] used *Pseudomonas* sp. ChlD isolated from a contaminated soil to biodegrade CLP in solution. Rochaddi et al. [16] isolated 116 bacterial strains from an aquifer, of which only 12 were able to degrade CLP in solution. Most of the strains belonged to the genus *Bacillus*. Moreover, Ishag et al. [17] isolated from a pesticide-polluted soil three bacteria from the genus *Bacillus* as degraders for CLP. Shabbir et al. [18] conducted biodegradation studies in solution with three bacterial strains isolated from domestic sewage water, *Pseudomonas aeruginosa*, *Enterobacter ludwigii*, and *Enterobacter cloacae*, and Ahir et al. [19] used *Tistrella* sp. AUC10 isolated from an agricultural field.

In the case of soil, the number of published studies is much lower than that in solution. The dissipation of CLP in sterile and nonsterile soil was studied in the presence of *Serratia rubidaea* ABS 10, observing that CLP was completely degraded and more rapidly dissipated than in controls' test [20]. In another study, *Dyadobacter jiangsuensis* 12851 isolated from an explosive-contaminated site showed a degradation of 76.93% after 30 d of inoculation [21]. Continuing in the same line of studies, *Pseudomonas* sp. was isolated from an industrial sewer and was able to remove up to 60 mg of CLP per kg of soil [22].

An essential factor in the effectiveness of pollutants biodegradation in soils is their availability [23]. In the case of CLP, its half-life is within 60 to 120 d in soil, but several authors have demonstrated that it could be increased to 1 year depending on environmental conditions [2,7]. CLP is highly hydrophobic (Log $K_{ow}$ 4.7), being a very persistent pesticide [24]. Cyclodextrins (CDs) have been recognised as an ecofriendly alternative to synthetic surfactants or organic solvents in order to increase contaminants availability in soils [25]. For this reason, CDs have been proposed as an option for the removal of pesticides present in soils because they can increase the water solubility of hydrophobic organic compounds. Few authors have used CDs as a pesticide enhancer bioavailability to accelerate their biodegradation. An improvement in diuron mineralisation was observed when hydroxypropyl-β-cyclodextrin (HPBCD) was employed [26]. RAMEB (randomly methylated-β-cyclodextrin) has also been used to improve the bioavailability and biodegradation of polychlorinated biphenyls (PCBs) in soil [27]. In another work, Rubio-Bellido et al. [28], showed HPBCD was an efficient tool for diuron mineralization in contaminated soils, reaching an improvement of the mineralization rate. However, only Báez et al. [29] studied the effect of different CDs on CLP soil adsorption–desorption equilibrium, showing a higher affinity between CLP and β-cyclodextrin (BCD). Báez et al. [30]

studied CLP biodegradation, observing a positive effect on the total microbial activity in the presence of BCD when dehydrogenase activity and fluorescein diacetate hydrolysis test was studied. Therefore, our study brings a novelty to the field of CLP degradation, combining the application of individual bacterial strains and CDs.

In this work, the bacterial strains *B. megaterium* CCLP1 or *B. safensis* CCLP2 were isolated in our laboratory from two agricultural soils treated with CLP for years, using enrichment cultures. They were inoculated in aqueous solution and in a soil spiked with CLP. Different biodegradation treatments were conducted: biostimulation (nutrient solution), bioaugmentation (isolated bacterial strains), and CDs. Finally, to check the viability of the decontamination strategy, ecotoxicological studies were performed to compare the state of the soil before and after the CLP bioremediation treatments.

## 2. Materials and Methods

### 2.1. Chlorpyrifos, Cyclodextrins, and Soils

Analytical grade (99%) chlorpyrifos [O, O-diethyl O-(3,5,6-trichloro-2-pyridyl) phosphorothioate] was provided from Sigma-Aldrich. Radiolabelled [ring-$^{14}$C]-CLP (41.35 mCi mmol$^{-1}$, purity 96.03%, and radiochemical purity 98.53%) was obtained from the Institute of Isotopes, (Budapest, Hungary. The CDs used (beta-cyclodextrin (BCD), HPBCD and RAMEB) were purchased from Cyclolab (Budapest, Hungary).

Five soil samples (ALC, LL, CR, PLD, R, Table 1) from the south of Spain were used. The ALC soil is located at Alcornocales Natural Park (36°20′54″ N, 5°36′14″ O); this soil is characterised by its high organic matter (OM) content (13.9%). The LL soil from Vejer de la Frontera-Cádiz (36°17′52.6″ N, 5°52′45.2″ W) is devoted to intensive agriculture. Organophosphate pesticides have been applied in this soil for many years. The CR soil was taken from the experimental farm La HAMPA (37°17′28.3″ N, 6°3′55.4″ W), which belongs to the Institute of Natural Resources and Agrobiology of Seville (IRNAS), in an area of olives where organophosphorus pesticides have not applied. The PLD soil originated from a crop of wheat, cereals, and vineyard located in Los Palacios y Villafranca-Seville (37°10′20.0″ N, 5°55′21.9″ W), and it has received the application of various organohalogen herbicides for years. From a palm trees area in Conil de la Frontera-Cádiz (36°18′32.4″ N, 6°08′58.6″ W) treated with a huge amount of CLP was collected the R soil. Soil samples were taken from the superficial horizon (0–20 cm) and were air-dried for 24 h and sieved (2 mm). Their physicochemical properties are shown in Table 1 and negligible amounts of CLP were detected in soils. The pH was determined in a proportion of 1 g:2.5 mL soil/water extract. The particle size distribution was evaluated using a Bouyoucos densimeter; the calcination or muffling method consisted in estimating the OM of the soil weight loss on ignition (LOI) or calcination, the quantification of organic matter was determined by $K_2Cr_2O_7$ oxidation, and manometric method was used to measure the total carbonate content.

**Table 1.** Some properties of the soils used.

| Soils | pH | $CO_3{}^{-2}$ N(%) | OM * (%) | Water Holding Capacity (%) | Sand (%) | Silt (%) | Clay (%) | Textural Classification | Taxonomic Classification ** |
|---|---|---|---|---|---|---|---|---|---|
| ALC | 5.1 | 0.5 | 13.9 | 73.4 | 69.1 | 7.8 | 23.1 | Sandy loam | Vertisol |
| R | 7.7 | 4.0 | 3.4 | 53.2 | 77.0 | 9.5 | 13.5 | Sandy | Alfisol |
| CR | 8.7 | 11.6 | 0.7 | 47.3 | 73.9 | 16.1 | 10.0 | Sandy | Inceptisol |
| PLD | 8.2 | 9.7 | 1.7 | 23.2 | 47.0 | 18.5 | 34.5 | Clay loam | Inceptisol |
| LL | 7.8 | 4.0 | 0.9 | 52.4 | 79.6 | 9.3 | 11.1 | Sandy | Alfisol |

* OM: organic matter. ** USDA SOIL TAXONOMY, soil maps, 2005. National Geographic Institute. Nature Database (Spanish Ministry of Environment).

### 2.2. Phase Solubility Studies

The solubility studies of CLP in the presence of various CDs were carried out based on the experiments reported by Higuchi and Connors, [31]. First, 5 mg of CLP was added to 20 mL of aqueous solutions (well above its water solubility, 2 mg L$^{-1}$) that contained different amounts of CDs (0−0.012 M for BCD and 0−0.1 M for HPBCD and RAMEB). The flasks were agitated at 25 °C for 7 d. Later, the suspensions were filtered through a 0.22 μm

Millipore glass-fibre membrane, and the concentration of dissolved CLP was determined. The concentration of supernatant was measured using gas chromatography (GC; Agilent GC 6890N) connected to a mass spectrometer (MS; Agilent MD 5975B) as described in the Analytical Methods Section 2.11. The apparent stability constants of the different CLP−CD complexes ($K_c$) were determined from the straight line obtained in the phase solubility diagrams according to the equation proposed by Higuchi and Connors [31].

$$Kc = \frac{slope}{S_0(1 - slope)} \tag{1}$$

where $S_0$ is defined as the CLP equilibrium concentration in aqueous solution when no CDs are present, and *slope* refers to the slope of the phase solubility diagram. Another parameter is the solubilisation efficiency ($S_e$), which is defined as the increment of CLP apparent solubility at the highest CD concentration studied regarding its water solubility.

### 2.3. CLP Mineralisation in Soils

The mineralisation studies of $^{14}$C-ring-labeled CLP in the five soils studied (ALC, LL, CR, R, PLD) under slurry suspension condition (continuous shaking at 120 rpm) were performed (in triplicate) by monitoring the evolution of produced $^{14}CO_2$, with the aim of revealing the potential capacity of the soil endogenous microbiota to degrade CLP. All the microcosm components were sterilised by autoclaving (Matachana steam steriliser model S100 with one cycle at 120 °C, pressure of 101 kPa, for 20 min), except the selected soil. The mineralisation tests were performed in respirometers which consist in a modified 250 mL Erlenmeyer flask with a soda tramp.

A quantity of 10 g of soil was spiked with a mixture of $^{14}$C-ring-labelled (450 Bq per flask) and unlabelled to obtain a final concentration of 50 mg kg$^{-1}$. For it, 0.25 mL of a 2000 mg L$^{-1}$ CLP stock solution in acetone, which also contained $^{14}$C-labelled CLP (450 Bq), was initially added to 2.5 g of soil (25% of the total soil) and was kept at room temperature under the fume hood for 24 h, the time necessary to evaporate completely the acetone. The remaining soil (75%) was then added and mixed, to avoid damage to the indigenous microbiota soil. Then, 50 mL of mineral salts medium (MSM) (which provided the macronutrients (g L$^{-1}$): Na$_2$HPO$_4$, 4.0; KH$_2$PO$_4$, 2.0; MgSO$_4$, 0.8; NH$_4$SO$_4$, 0.8) were added. A total of 1 mL of micronutrients (SNs: NiCL$_2$ 6H$_2$O, 12.5; SnCl$_2$ 2H$_2$O, 25.0; ZnSO$_4$ 7H$_2$O, 12.5; Al$_2$(SO$_4$)$_3$ 18H$_2$O, 12.5; MnCl$_2$ 4H$_2$O, 75.0; CoCl$_2$ 2H$_2$O, 12.5; FeSO$_4$ 7H$_2$O, 37.5; CaSO$_4$ 2H$_2$O, 10; KBr, 3.75; KCl, 3.75; LiCl, 2.5 (mg L$^{-1}$) [13]) was also added and the Erlenmeyer flasks were closed with Teflon-lined stoppers before incubation at 30 ± 1 °C for 100 days. The mixture of MSM and SNs (50:1) was named nutrients solution (NS). $^{14}CO_2$ was trapped in the alkali trap of the biometer flask and measured as the radioactivity appearing in the alkali trap by extracting periodically the NaOH solution and mixing it with 3 mL of a liquid scintillation cocktail (Ready safe from PerkinElmer, Inc., Waltham, MA, USA). This mixture was stored in darkness for about 24 h with the aim of dissipating the chemiluminescence. Radioactivity was evaluated using a liquid scintillation counter (Beckman Instruments Inc., Fullerton, CA, USA, model L55000TD).

### 2.4. Chlorpyrifos Microbial Degrader Isolation by Enrichment Culture

Soils that had shown natural capacity to mineralise CLP (LL and R) were selected to carry out enrichment cultures to isolate potential CLP degrading strains. A quantity of 10 g of each soil was added to sterilised 250 mL Erlenmeyer flasks with 50 mL of MSM spiked with 1 g L$^{-1}$ of CLP as the only source of carbon and energy. Then, 1 mL of SNs was added to the MSM solution. The incubation conditions of the orbital shaking culture were 170 rpm at 30 °C, and every week (4 times) 10 mL of the culture was removed and transferred to another Erlenmeyer flask containing 40 mL of nutrients solution (MSM + SNs) in the presence of the contaminant and it was incubated again for 7 d. Aliquots of 100 μL of the final enrichment cultures were spread on agar plates prepared with MSM medium and CLP with a concentration of 0.02 g L$^{-1}$ according to Alley and Brown [32]. Successive isolations

were performed recognising and selecting different colonies according to macroscopic features, such as their size, colour, edge, and elevation. In total, 15 and 11 strains were isolated from the R and LL soil, respectively. The selected isolated strains were preserved in Eppendorf with a 40% solution of glycerol, and they were kept at $-80$ °C.

### 2.5. Chlorpyrifos-Degrading Capacity of the Isolated Strains

The capacity of the isolated strains to remove CLP was tested through a preliminary degradation experiment in solution. A bacterial culture with a density of approximately $10^8$ CFU (colony forming units) per mL (optical density, $OD_{600} = 1$), and a CLP concentration of 10 mg $L^{-1}$ in MSM + SNs (50:1) was added to glass bottles. The samples were incubated in a thermostatic chamber at 30 °C for 20 d with agitation, and the final concentration of CLP in solution was measured by GC–MS as described in the Analytical Methods Section 2.11.

### 2.6. Degrading Strain Identification by 16S rDNA Amplification

The degrading strains isolated from the LL and R soils that showed the best CLP degradation capacity (Figure S1) were selected to carry out more complete CLP biodegradation assays in solution. An aliquot of an LB culture of a degrading bacterium was centrifuged (11,000 rpm, 1 min) and then the obtained pellet was used to extract its DNA using the G-spinTM total DNA Extraction Kit (iNtRON Biotechnology). The 16S rRNA gene was amplified by polymerase chain reaction (PCR) using a high-fidelity polymerase (Velocity DNA polymerase from Bioline) with universal oligonucleotides primers: 16F27 (annealing at position 8–27 *E. coli* numbering) and 16R1488 [33]. Eventually, the PCR products were purified using PCR clean-up Gel Extraction kit NucleoSpin® Gel and PCR clean-up (Macherey-Nagel) to be sent for sequencing.

### 2.7. Chlorpyrifos Biodegradation Experiments in Solution

Biodegradation experiments of CLP insecticide were conducted in 25 mL sterilised glass vials in triplicate. Each vial contained the bacterial inoculum required to reach a final density of $10^8$ CFU $mL^{-1}$ of LLCCLP4 or RCCLP11.

Quantities of 15 µL of NS and 15 mL of MSM were used, contaminated with 10 mg $L^{-1}$ of CLP as the only source of energy and carbon. Uninoculated vials were used to control abiotic degradation. The vials were located at 30 °C in an incubator–shaker (150 rpm) for 60 d. Different samples were taken at initial time and after pre-established periods of time (1, 3, 7, 12, 21, 30, and 60 d) to monitor the removal of the investigated pesticide. Samples were taken in a vertical laminar flow cabin, centrifuged (7000 rpm, 20 min), and a supernatant aliquot was kept in 1.5 mL glass vials. CLP was quantified by high-performance liquid chromatography (HPLC) as described below.

The enumeration of colony forming units per gram of soil (CFU $g^{-1}$ soil) for the total number of CLP-degrading microorganisms in the ALC soil were counted using the spread plate technique on petri dishes with MSM agar supplemented with 50 mg $L^{-1}$ of CLP. Then, 1 g of soil was extracted with 5 mL of MSM, and then 100 µL of the extract serially diluted (1:10). Aliquots (100 µL) of the resultant solutions were spread over agar plates and incubated at 30 °C, with plate counts conducted at 7 d.

### 2.8. Chlorpyrifos Biodegradation Experiments in Soils

The biodegradation tests of CLP in soils were conducted in 25 mL sterilised glass vials, containing 1 g of the soil ALC spiked with 50 mg $kg^{-1}$ CLP and the necessary volume of MSM and SNs to reach 40% of the soil water holding capacity (WHC, 73.44%). Several biodegradation strategies were designed: (i) biostimulation (contaminated soil sample + NS), which was used as a control of the activity of the endogenous soil microbiota; (ii) bioaugmentation (contaminated soil sample + NS + *B. megaterium* CCLP1 or *B. safensis* CCLP2), where the soil sample was inoculated with $1 \times 10^8$ CFU $g^{-1}$; (iii) the addition of RAMEB solution (contaminated soil sample + NS + RAMEB), where RAMEB was added with an amount corresponding to 10 times that of the CLP molar concentration initially

added in the soil sample; and (iv) a combined use of biostimulation, bioaugmentation, and RAMEB (contaminated soil sample + NS + RAMEB + *B. megaterium* CCLP1 or *B. safensis* CCLP2). In parallel, abiotic CLP degradation controls were performed by adding 200 mg L$^{-1}$ of HgCl$_2$. All experiments were kept at 30 °C in a laboratory incubator for 100 d. Samples were taken at different times of the incubation (0, 1, 4, 7, 14, 21, 42, 60, 89, and 100 d). Residual CLP was measured in the soil samples. Briefly, 1 g of soil sample was extracted with 5 mL of acetonitrile/water (90:10). The extraction process was carried out through the following steps: (1) 1 min of vortex mixer, (2) 10 min in an ultrasound bath, (3) 1 h of agitation of the tubes in an orbital shaker at 100 rpm at 20 ± 1 °C, and (4) 10 min centrifugation at 8000 rpm. The CLP concentration in the supernatant, after filtering through a 0.22 μm Millipore glass fibre membrane, was measured by HPLC as pointed out in the section of Analytical Methods 2.11.

### 2.9. Biodegradation Kinetic Modelling

Biodegradation curves were fitted to the most appropriate kinetic model, using an Excel file provided by the FOCUS [34] workgroup on degradation kinetic. This program uses the solver tool (Microsoft statistical package) and rate curves. Curves were fitted to three first-order kinetic models: a simple first-order model (SFO) and a biphasic first-order sequential model (hockey stick, HS) and a first-order multicompartment model (FOMC) according to the following equations:

$[C]_t = [C]_0\, e^{-kt}$ (SFO)

$[C]_t = [C]_0\, e^{-k1tb}\, e^{-k2(t-tb)}$ (HS)

$[C]_t = M_0/((t/\beta)\,1)\,\alpha$ (FOMC)

$DT_{50} = \ln 2/k$ (SFO)

$DT_{50} = (\ln 100/100 - 50)\,/\,k_1$ if $DT_{50} \leq tb$ (HS)

$DT_{50} = tb + (\ln (100/100 - 50) - k_1\, tb)\,/\,k_2$ if $DT_{50} > tb$ (HS)

$DT_{50} = \beta\,(2\,(1/\alpha) - 1)$ (FOMC)

$[C]_t$: concentration of biodegradation at time t.

$[C]_0$: concentration of biodegradation at the beginning.

$k_1$, $k_2$: rate constants of biodegradation (d$^{-1}$).

$DT_{50}$: required time for the pollutant concentration to decline to half of its initial value.

tb: time at which a change in the rate constant is observed.

α: shape parameter determined by the coefficient of variation of k values.

B: location parameter

The SFO, HS, and FOMC models were chosen for their relatively simplicity, but they have potential to adjust the measured dissipation kinetic datasets for monophasic or biphasic biodegradation [35]. The Chi-squared ($\chi^2$) test was used to estimate the appropriateness of the model and to assess the accuracy of each resulting fit. This test considers the deviations between observed and calculated values (numerator) for each model in relation to the uncertainty of the measurements (denominator). To assess the goodness of fit of the degradation kinetics models to the experimental data, the best fit with the lowest $\chi^2$ and scaled error values were considered.

### 2.10. Chlorpyrifos Availability in Soil

CLP extraction, from samples of contaminated soil, was conducted to verify the effect of using NS and RAMEB as extractants on CLP availability. Corex glass centrifuge tubes containing 1 g of the soil sample contaminated with CLP (50 mg kg$^{-1}$) were extracted with 5 mL of NS or NS combined with RAMEB (10 times the molar concentration of CLP initially added in soil). Tubes were shaken in an orbital shaker for 72 h at 20 ± 1 °C, and centrifuged (10 min, 7000 rpm), then supernatant was filtered using a 0.22 μm Millipore glass fibre membrane. The CLP concentration was monitored using the analytical method described in Section 2.11.

### 2.11. Chlorpyrifos Analytical Method

The samples obtained in CLP were analysed by high-performance liquid chromatography (HPLC), using a Varian ProStar 410 HPLC AutoSampler equipment, a Kromasil C18 reverse-phase column ($15 \times 0.40$ cm$^2$), and the mobile phase consisted of a mixture of acetic acid glacial/water/acetonitrile ($0.1\ v$/$10\ v$/$90\ v$), with a flow rate of 1 mL min$^{-1}$ and an injection volume of 20 µL, at λ of 290 nm for the detection of CLP, at a retention time of 2.07 min.

### 2.12. Toxicity Analysis

The bioluminescence of the marine bacterium *Vibrio fischeri* was employed in the Microtox® Test System to measure the toxicity of CLP in solution and soil systems, based on the standard protocol using the basic test (UNE-EN ISO 11348-3/A1:2019). Samples from the CLP biodegradation assays in solution using *B. megaterium* CCLP1 and *B. safensis* CCLP2 were centrifuged for 10 min to 7000 rpm and were serially diluted (1:2) with 2% NaCl solution. In the case of soil samples, 2 g of soil sample was added to 3 mL of 2% NaCl solution. These suspensions were shaken for 10 min, centrifuged (2 min, 10,000 rpm) and serially diluted (1:2) with 2% NaCl solution, according to [36]. *V. fischeri* bacteria were rehydrated immediately prior to use. Assays were conducted in a temperature-controlled photometer at 15 °C (Microbics Corporation (1992). Both kinds of samples were measured at the beginning and 60 d after inoculation and compared with the control.

The EC$_{50}$ parameter (soil extract concentration (% $v$/$v$) having a toxic effect on 50% of *V. fischeri*) was given by the Microtox® Text System for each sample analysed. The EC$_{50}$ value corresponds to the CLP concentration (% $v$/$v$) having a toxic effect on 50% of the bacterial population. Toxicity values were then expressed in toxic units (TU), using the formula TU = 100/EC$_{50}$. TU results were classified according to Persoone et al. [37]: TU < 0.4, no acute toxicity; 0.4 < TU < 1, light acute toxicity; 1 < TU < 10, acute toxicity; 10 < TU < 100, high acute toxicity; TU > 100, very high acute toxicity.

## 3. Results

### 3.1. Chlorpyrifos Mineralisation in Selected Soils

Five soils were artificially contaminated with 50 mg kg$^{-1}$ of CLP. This concentration was selected to simulate a point-source contamination by pesticides, such as an accidental pesticide spill or the accumulation of pesticides through their repeated application [26,38]. Figure 1 shows the CLP mineralisation curves for the different soils studied, obtained by the action of their endogenous microbiota, which was stimulated in the presence of SNs + MSM. In the control assays (without SNs), mineralisation was not observed (data not shown).

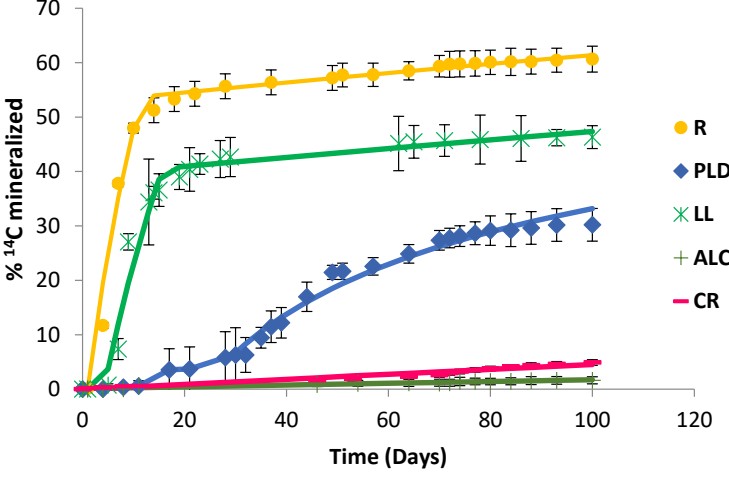

**Figure 1.** Chlorpyrifos mineralisation curves (100 d) in R, PLD, LL, ALC, and CR soils.

In the studies with the ALC and CR soils, the mineralisation curves only reached 0.7% and 4.5% of the initial CLP content after 100 d of testing, respectively (Table 2).

**Table 2.** Kinetic parameters calculated from chlorpyrifos mineralisation curves obtained under natural attenuation conditions for the studied soils.

| Soil | Kinetic Model | $K_1$ (d$^{-1}$) | $K_2$ (d$^{-1}$) | tb (d) | $\alpha$ (d$^{-1}$) | $\beta$ (d$^{-1}$) | DT$_{50}$ (d) * | % Extension of Mineralisation | Lag Phase (d) ** | $\chi^2$ *** | Scaled Error | R$^2$ |
|------|--------------|------|------|------|------|------|------|------|------|------|------|------|
| LL | HS | $4.5 \times 10^{-2}$ | $1.4 \times 10^{-3}$ | 15.7 | - | - | 1573 | 47.3 | 5 | 1.29 | 0.29 | 0.97 |
| PLD | FOMC | - | - | - | 0.3 | 0.4 | 485 | 33.2 | 11 | 0.5 | 1.21 | 0.97 |
| R | HS | $7.2 \times 10^{-2}$ | $2.0 \times 10^{-3}$ | 11.7 | - | - | 9.6 | 61.4 | 4 | 1.63 | 0.16 | 0.98 |
| ALC | SFO | $1.6 \times 10^{-4}$ | - | - | - | - | 4203 | 0.7 | - | 0.003 | 0.10 | 0.95 |
| CR | SFO | $4.7 \times 10^{-4}$ | - | - | - | - | 1474 | 4.5 | - | 0.02 | 0.29 | 0.95 |

\* DT$_{50}$: time to decline to half the initial concentration of CLP. \*\* Acclimatisation phase was not observed (-). \*\*\* $\chi^2$ calculated values < $\chi^2$ corresponding tabulated value (*p* 0.05).

Moreover, kinetic modelling calculated that 50% of mineralisation (DT$_{50}$) would be reached after 4203 d in the case of the ALC soil and 1474 d for the CR soil, periods too long for a remediation process. In the case of the PLD soil, an acclimatisation period of 11 d was required prior to removing CLP from the soil (33.2% mineralised and DT$_{50}$ 485 d). An increase in the percentage of mineralisation was observed for the LL and R soils (47.3 and 61.4% mineralised, respectively), and the degradation activity of the microbiota in the presence of CLP began to be observed after 5 and 4 d, respectively.

### 3.2. Isolation and Characterisation of the Potential Chlorpyrifos-Degrading Bacteria from Selected Soils

For the isolation of CLP-degrading bacteria, those soils that presented an endogenous microbiota capable of mineralising a high percentage of CLP were used (LL and R). In total, 11 bacterial strains were isolated from the LL soil and 15 from the R soil. A preliminary biodegradation test in solution was conducted by monitoring the CLP concentration at the beginning and after 20 d of assay. Figure S1 shows those bacterial strains which exhibited CLP degradation. In the case of the LL soil, only one strain, LLCLP4, showed a significant capacity to degrade CLP in solution, achieving an extent of biodegradation of 70.3% (Figure S1A). In the case of the R soil, eight bacterial strains were able to biodegrade different CLP percentages (12.8, 63.8, 35.1, 66.8, 50.5, 26.3, 18.4, and 59.4%) after 20 d (Figure S1B), respectively. LLCLP4 and RCLP11 were identified and selected to conduct biodegradation assays due to their capacity to use CLP as the only source of carbon and energy quite effectively. The 16S rRNA gene sequence showed a match of 100% to species from *Bacillus megaterium* and *Bacillus safensis*, respectively, in the NCBI GenBank, and they are called from now on *Bacillus megaterium* CCLP1 and *Bacillus safensis* CCLP2 (GenBank accession number: MT293409 and MT293353, respectively).

### 3.3. Chlorpyrifos Biodegradation in Solution by B. megaterium CCLP1 and B. safensis CCLP2

The two bacterial strains previously identified as *B. megaterium* CCLP1 and *B. safensis* CCLP2 were selected to carry out CLP biodegradation and mineralisation assays in solution by bioaugmentation. Both bacterial strains failed to mineralise CLP (data not shown), and biodegradation results are presented in Figure 2. The curves obtained by quantifying nonbiodegraded CLP were adjusted to the single first-order kinetic model (SFO). The kinetic parameters calculated from the biodegradation curves obtained in the presence of the selected bacterial strains are shown in Table 3. There was no evidence of CLP abiotic dissipation in the uninoculated control. For both bacterial strains a significant percentage of biotransformation was observed, reaching a degradation calculated according to the model of about 99.1% for *B. megaterium* CCLP1, while in the case of *B. safensis* CCLP2, it was 98.9% after 60 d of assay (Table 3). Both bacterial strains, belonging to the same genus (*Bacillus*), showed a very similar biodegradation profile, reaching 50% of biodegradation after about 9 d of assay.

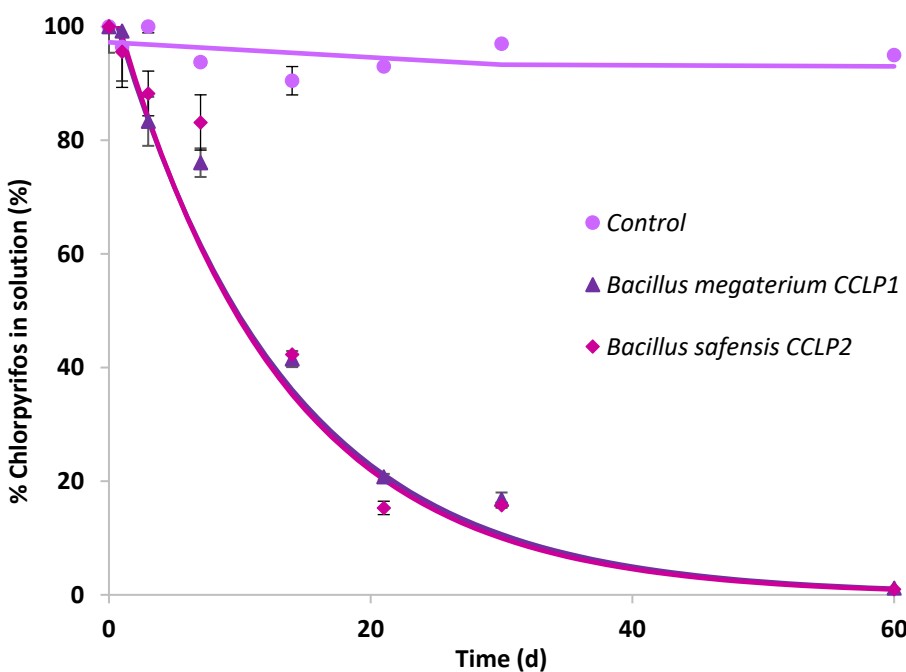

**Figure 2.** Chlorpyrifos biodegradation curves in solution after inoculation of *B. megaterium* CCLP1 (▲) or *B. safensis* CCLP2 (◆). Solid lines show model fitting to the experimental results (symbols).

**Table 3.** Kinetic parameters calculated from the chlorpyrifos biodegradation curves in solution after inoculation with *B. megaterium* CCLP1 and *B. safensis* CCLP2.

| Treatment | $K_1$ (d$^{-1}$) | DT$_{50}$ (d) * | Extent of Biodegradation (%) | $\chi^2$ ** | Scaled Error | $R^2$ |
|---|---|---|---|---|---|---|
| *B. megaterium* CCLP1 | $7.6 \times 10^{-2}$ | 9.1 | 99.1 | 14.5 | 6.98 | 0.94 |
| *B. safensis* CCLP1 | $7.9 \times 10^{-2}$ | 8.8 | 98.9 | 13.4 | 6.3 | 0.95 |

* DT$_{50}$: time to decline to half the initial concentration of CLP. ** $\chi^2$ calculated values < $\chi^2$ corresponding tabulated value (*p*: 0.05). Biodegradation curves were fitted to a single first-order kinetic model.

### 3.4. Chlorpyrifos Biodegradation in Soil by B. megaterium CCLP1 and B. safensis CCLP2

Both strains *B. megaterium* CCLP1 and *B. safensis* CCLP2 were also selected for performing CLP biodegradation experiments in soil. The ALC soil (Table 1), from a natural park, was used to carry out biodegradation studies, since its microbiota did not present CLP degrading microorganisms, and in addition, the ALC soil presented the particularity that its OM content was extremely high (13.9%). This soil did not show CLP mineralisation capacity after 100 d when NS was added as a biostimulant of its endogenous microbiota (Figure 1). To quantify the potential degrading capacity of this endogenous microbiota, the potential CLP-degrading CFU were determined in petri dishes with MSM media in the presence of 50 mg L$^{-1}$ of CLP as described in the Materials and Method section. A value of $8.6 \times 10^2$ CFU g$^{-1}$ of ALC soil was reached, concluding that the number of microorganisms with degradative capacity in this soil was low. The result was in line with the origin of the studied soil. For the CLP biodegradation studies, a bioaugmentation treatment would be necessary to increase the chances of successful remediation. Figure 3 shows the different biodegradation curves obtained from biodegradation assays. The CLP abiotic removal was evaluated, where a HgCl$_2$ (200 mg L$^{-1}$) solution was added to the soil with the aim of eliminating the soil endogenous microbiota, discarding abiotic processes (figure not shown). The role of endogenous soil microbiota in the CLP biodegradation was evaluated (control), but no significant biodegradation was observed in the investigated soil. However, when the microbiota was stimulated using NS, an increase in the biodegradation percentage was observed (15.7%), with a DT$_{50}$ value of 697 d (almost 2 years), indicating the extremely high persistence of CLP in this soil, and the low capacity of its microbiota

for CLP degradation, as previously indicated. This result demonstrated the need to apply biological techniques (bioaugmutation) together with NS (biostimulation) to improve the extent of the biodegradation, as well as to accelerate the rate of biodegradation. For this purpose, *B. megaterium* CCLP1 and *B. safensis* CCLP2 were inoculated individually, obtaining biodegradation curves that were adjusted to the HS first-order kinetic model (Table 4). Percentages of 60.6% and 64.8% of CLP biodegradation were achieved after 100 d of treatment, respectively. It should be noted that $DT_{50}$ radically decreased from 697 d (biostimulation) to 44.6 and 47.1 d in the case of *B. megaterium* CCLP1 and *B. safensis* CCLP2, respectively. In both cases, the bacterial strains followed a similar behaviour, which may be because both belong to the same bacterial genus.

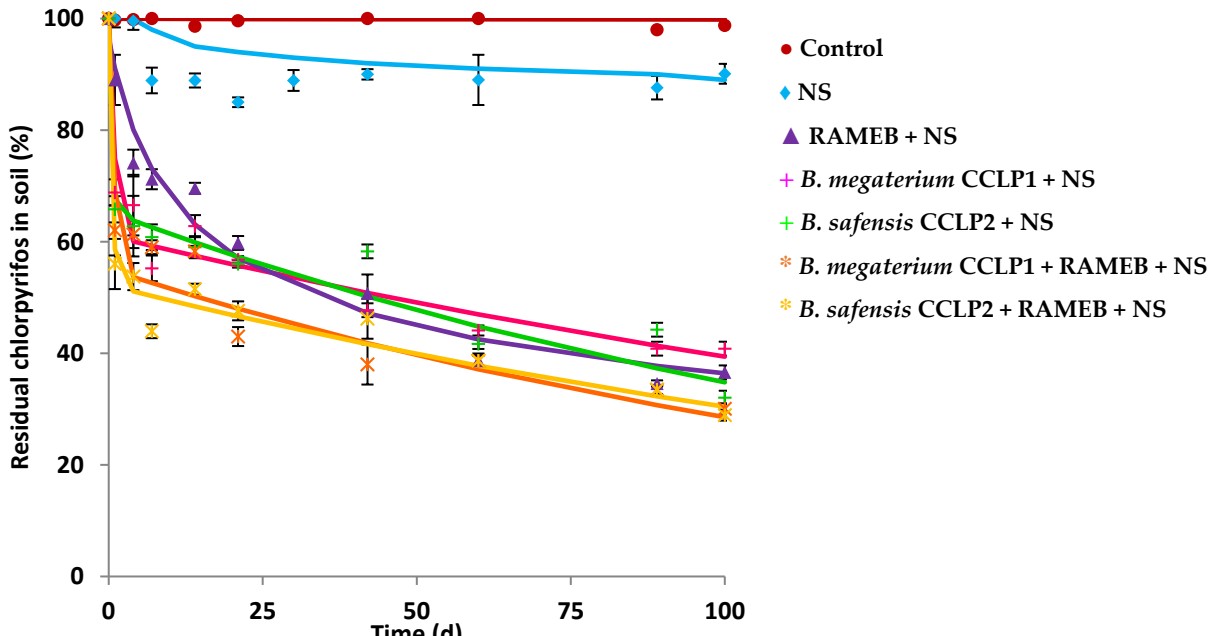

**Figure 3.** Chlorpyrifos biodegradation curves in ALC soil after the application of: NS (♦), RAMEB + NS (▲), *B. megaterium* CCLP1 + NS (+), *B. safensis* CCLP2 + NS (+), *B. megaterium* CCLP1 + RAMEB + NS (✳), *B. safensis* CCLP2 + RAMEB + NS (✳), and control (●). Solid lines show model fitting to the experimental results (symbols).

**Table 4.** Kinetic parameters calculated from chlorpyrifos biodegradation curves after different bioremediation treatments were applied on ALC-contaminated soil.

| Treatment | Kinetic Model | $K_1$ ($d^{-1}$) | $K_2$ ($d^{-1}$) | tb (d) | α ($d^{-1}$) | β ($d^{-1}$) | $DT_{50}$ (d) * | $DT_{90}$ (d) ** | Extent of Biodegradation (%) | $x^2$ *** | Scaled Error | $R^2$ |
|---|---|---|---|---|---|---|---|---|---|---|---|---|
| NS | SFO | $9.9 \times 10^{-4}$ | - | - | - | - | 697 | 6454.4 | 15.7 | 0.64 | 1.90 | 0.91 |
| RAMEB | FOMC | - | - | - | 0.3 | 5.1 | 38.9 | 2314 | 63.6 | 2.35 | 3.19 | 0.96 |
| *B. megaterium* CCLP1 | HS | 0.3 | $4.3 \times 10^{-3}$ | 1.7 | - | - | 44.6 | 411.4 | 60.6 | 2.35 | 0.76 | 0.95 |
| *B. safensis* CCLP2 | HS | $3.6 \times 10^{-1}$ | $6.3 \times 10^{-3}$ | 1.1 | - | - | 47.1 | 397.5 | 64.8 | 3.03 | 1.41 | 0.95 |
| *B. megaterium* CCLP1 + RAMEB | HS | 0.4 | $6.5 \times 10^{-3}$ | 1.6 | - | - | 14 | 258.9 | 71.5 | 10.16 | 0.76 | 0.88 |
| *B. safensis* CCLP2 + RAMEB | HS | $5.4 \times 10^{-1}$ | $5.4 \times 10^{-3}$ | 1.2 | - | - | 7.9 | 305.9 | 69.6 | 1.68 | 0.90 | 0.97 |

* $DT_{50}$: time to decline to half the initial concentration of CLP. ** $DT_{90}$: time to decline to 90% of the initial concentration of CLP. *** $x^2$ calculated values < $x^2$ corresponding tabulated value (*p*: 0.05).

### 3.5. Chlorpyrifos Solubility in the Aqueous Phase in the Presence of Different Cyclodextrins

As was mentioned, CLP bioavailability can be limited due to its low water solubility (1.05 mg $L^{-1}$) and high hydrophobicity (log $k_{ow}$ 4.7), which means an important persistence of the chemical in soil, due to its high adsorption on soil OM. Since the OM of the selected soil was very high (13.9%), this study proposes the use of CDs to achieve an increase in water solubility of the contaminant accelerating its biodegradation in the soil solution. Hence, three CDs were studied as bioavailability enhancers and their phase solubility diagrams are shown in Figure 4.

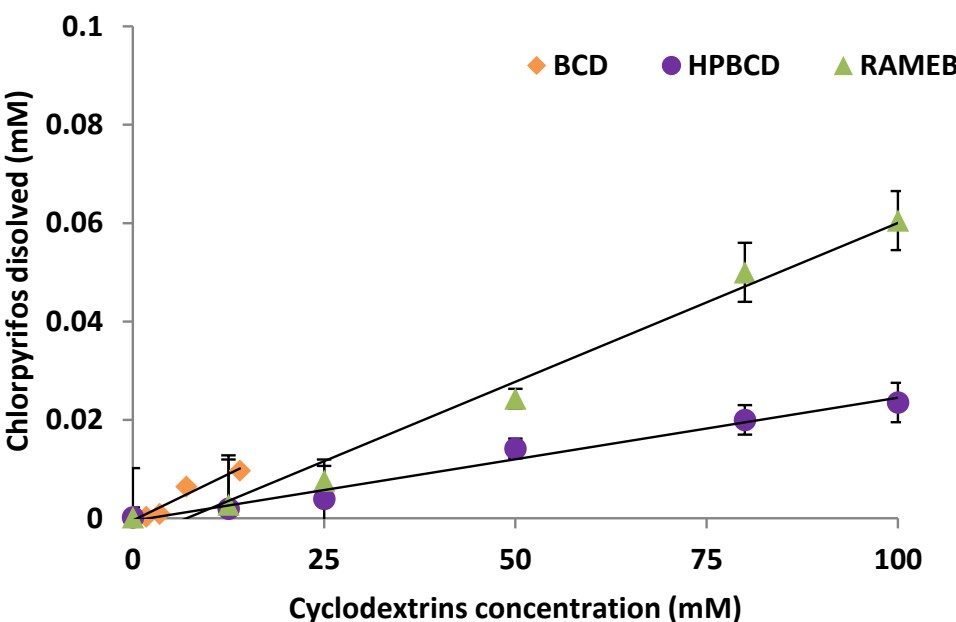

**Figure 4.** Chlorpyrifos phase solubility diagrams in the presence of the cyclodextrins studied.

A linear enhancement in CLP water solubility was observed as the CD concentration increased, but a solubility threshold could not be reached, indicating an inclusion complex formation with a 1:1 stoichiometry (slope < 1) [39]. It involves $A_L$ diagrams according to Higuchi and Connors [31]. As far as we know, there are not many antecedents for CLP regarding the formation of complexes with CDs, apart from the studies carried out by Báez et al. [29].

$K_c$ (apparent stability constant) was calculated from the slope of the straight lines in the diagram using the equation proposed in the Materials and Methods section, and $S_e$ (solubility efficiency) corresponding to the increase in solubility calculated for the highest CD concentration studied referred to the CLP water solubility in the absence of CD ($S_0$) [40]. The $K_c$ and $S_e$ values calculated in the presence of the different CDs are shown in Table 5. The best $S_e$ was reached when RAMEB was studied, multiplying by 311 the solubility of CLP in water.

**Table 5.** Chlorpyrifos apparent stability constants ($K_c$) and solubilisation efficiency ($S_e$) obtained from the phase solubility diagrams.

|  | $k_c$ (M$^{-1}$) | $S_e$ [a] | $R^2$ |
|---|---|---|---|
| RAMEB | 3090 ($\pm$ 1.2) | 311 ($\pm$ 1.1) | 0.98 |
| HPBCD | 1545 ($\pm$ 1.1) | 121 ($\pm$ 1.4) | 0.97 |
| BCD | 4119 ($\pm$ 0.9) | 50.2 ($\pm$ 0.9) | 0.96 |

[a] Increment of CLP solubility for the highest CD concentration used with respect to CLP solubility in the absence of CD.

### 3.6. Chlorpyrifos Biodegradation in Soil Assisted by Cyclodextrin

RAMEB was applied to ALC-contaminated soil (Figure 3) causing an improvement in the extent of the CLP biodegradation (63.6%, Table 4), and a decrease in DT$_{50}$ (38.9 d), regarding the treatment with only NS (15.7%, DT$_{50}$ = 696.6 d). Soil CLP extraction experiments were conducted to test the effect of water, NS, and RAMEB solutions on the CLP availability on the studied soil. The results are shown in Figure S2. After 72 h in the presence of NS, the percentage of CLP extracted was low (0.2%), but when RAMEB was used as extractant, the extracted amount increased from 0.2% to 8.2%. That is, the percentage of CLP extracted was 41-fold higher with RAMEB than with the NS solution.

In the framework of this study, *B. megaterium* CCLP1 and *B. safensis* CCLP2 were combined with RAMEB, demonstrating to be the most effective strategy of bioremediation

(Figure 3). The biodegradation rate increased in comparison to previous treatments (up to 71.5% and 69.6%, respectively), and a notable improvement in biodegradation kinetic parameters was attained, decreasing $DT_{50}$ to 14 and 7.9 d, respectively (Table 4).

### 3.7. Toxicity of Chlorpyrifos in Aqueous and Soil Samples

The toxicity of the CLP and their metabolites formed during the biodegradation process was evaluated for aqueous (after 60 d) and soil (after 100 d) samples in the presence of individual bacterial strains (*B. megaterium* CCLP1 and *B. safensis* CCLP2) and/or RAMEB with the standardised Microtox® test. With this system it is possible to easily detect a reduction in the luminescence of the bacterium *Vibrio fischeri* as toxicity increases, and the results are well reproduced [41]. The toxicity parameters (TU and $EC_{50}$) were determined and are summarised in Table 6.

**Table 6.** Acute toxicity test towards *V. fischeri* after the testing time (after 60 d of incubation in solution, and after 100 d in ALC soil contaminated with chlorpyrifos) in the presence of *B. megaterium* CCLP1, *B. safensis* CCLP2, and/or RAMEB.

|  | Treatment | TU * | Toxicity ** |
|---|---|---|---|
| Solution | Without treatment | 2.4 | Acute |
|  | *B. megaterium* CCLP1 | 1.5 | Acute |
|  | *B. safensis* CCLP2 | 2.2 | Acute |
| Soil | Without treatment | 5.6 | Acute |
|  | *B. megaterium* CCLP1 | - | Nontoxic |
|  | *B. safensis* CCLP2 | 0.004 | Nontoxic |
|  | *B. megaterium* CCLP1 + RAMEB | - | Nontoxic |
|  | *B. safensis* CCLP2 + RAMEB | - | Nontoxic |

* *TU: toxic units.* ** According to Persoone et al. [37].

The initial concentration of CLP in solution (10 mg $L^{-1}$) was classified as "acute toxicity" (1 < TU < 10, acute toxicity) according to Persoone et al. [37], with a value of TU = 2.4. After inoculating *B. megaterium* CCLP1 and *B. safensis* CCLP2, TU was found to decrease only slightly in both cases to 1.5 and 2.2 TU, respectively, regarding the initial concentration.

Table 6 also shows the results of toxicity after the application of the treatments (bioaugmentation and/or RAMEB) in an artificially contaminated soil (50 mg $Kg^{-1}$). The TU at the beginning of the biodegradation experiment reached a value of 5.6, which indicated that the extract of the contaminated soil when no treatment was applied presented acute toxicity. When *B. megaterium* CCLP1 was inoculated in the soil, the toxicity was undetectable at the end of the treatment. Moreover, when *B. safensis* CCLP2 was employed, the level of toxicity decreased (TU = 0.004) from acute toxicity to nontoxic (TU < 0.4). By combining both techniques, bioaugmentation and RAMEB, the treatment showed its efficacy without detecting toxicity in any of the cases (Table 6).

### 4. Discussion

Based on the mineralisation results, the ALC, CR and PLD soils were discarded as sources of CLP-degrading microorganisms. However, an increase in the percentage of mineralisation was observed for LL and R, possibly due to the previous adaptation of the microorganisms present in the soil to the presence of CLP or compounds with similar chemical structures. The presence of pollutants in these soils due to their agricultural origin, could have produced an alteration of the native microbial communities, favouring the development of some microbial taxa capable of using them as a source of carbon and energy, while other microorganisms became less prevalent in contaminated soils [42]. Different studies conclude that pesticide removal is dependent on the repeated application of the compound in an agricultural soil, causing a rapid response of the endogenous microbiota

against pesticide [43,44]. For this reason, LL and R soils were selected to isolate potential CLP-degrading bacteria.

*B. megaterium* CCLP1 and *B. safensis* CCLP2, isolated from soils LL and R showed the best CLP degradation results in solution, being able to eliminate the initial concentration of CLP (10 mg L$^{-1}$), without the need to resort to the application of microbial consortia. This result is in line with the fact that *Bacillus* genus is frequently found in soils as a CLP degrader. Onwona-kwakye et al. [45] studied by a 16S rRNA analysis sequencing the changes caused in the endogenous microbiota of agricultural soils exposed to different pesticides including CLP. The frequency of the *Bacillus* genus increased in areas exposed to pesticides, highlighting that this bacterial genus could be useful for CLP bioremediation

Some works have described species of the genus *Bacillus* as CLP-degrading in solution, particularly *B. cereus*. Duraisamy et al. [46] employed *B. cereus* MCAS 02, isolated from an agricultural soil to degrade CLP at different agitation rates, pH, and yeast extract concentrations. Elshikh et al. [47] showed the degradation ability of *B. cereus* CP6 and *Klebsiella pneumoniae* CP19 isolated from wastewater sediment. Seven strains of *B. cereus* isolated from an aquifer were demonstrated to degrade CLP in solution [16]. Farhan et al. [48] also studied the efficacy of *B. cereus* Ct3, isolated from a contaminated agricultural soil. Eissa et al. [49] described another bacterial strain belonging to the *Bacillus* genus (*Bacillus* sp. SMF5) and *Streptomyces thermocarboxydus* A-B for CLP degradation in solution, and *B. pumilus*, isolated from a cotton soil, was used by Anwar et al. [50]. In relation to the two bacterial strains used in the present study, only Ishag et al. [17] had previously mentioned *B. safensis* as a CLP degrader, with 90% of CLP dissipated after 30 d in solution, similar to the results reached in the present paper, and Zhu et al. [51] mentioned the strain *B. megaterium* CM-Z19, but as a degrader of chlorpyrifos-methyl.

However, as far as we know, only one strain belonging to the *Bacillus* genus has been described as a CLP degrader in soil, which brings greater relevance to our study. Zhu et al. [52] isolated from an agricultural soil the strain *B. licheniformis* ZHU-1 with capacity to degrade CLP. Nevertheless, species belonging to other genera have been used individually or in consortia as CLP degraders in soils. *Naxibacter* sp. CY6 [53], *Stenotrophomonas* sp. YC-1 [54], *Pseudomonas putida* CBF10-2, *Ochrobactrum anthropic* FRAF13, and *Rhizobium radiobacter* GHKF11 were employed to form a bacterial consortium [55], or *Achromobacter xylosoxidans* JCp4 and *Ochrobactrum* sp. FCp1 inoculated together [56].

It is important to mention that CLP is strongly adsorbed by soils because of its low water solubility (1.05 mg L$^{-1}$) and high soil sorption capacity (Log $k_{ow}$ = 4.7), which may result in a high accumulation of CLP in soils (DT$_{50}$ = 386, very persistent) [24,57], making the removal of CLP from the soil difficult. Another aspect to highlight in the present study is the high OM content in the ALC soil, since previous studies have been performed in soils with much lower OM contents, or the effect of this important parameter has not been considered at all, and the information about the OM content is missing. CLP is strongly adsorbed on the OM of the soils due to its extremely high hydrophobicity, and it has been observed that as the OM of soils increases, the formation of CLP-bound residues increases, reducing its availability [57]. It is likely that for this reason, the biodegradation curves in the presence of degrading bacteria fitted to an HS kinetic model, where k1 and k2 were the rate constants of degradation for the fast and the slow fraction, respectively [34]. k1 showed a quick degradation and k2 a slower degradation, possibly due to a severe bioavailability decline of the insecticide. To improve bioremediation strategy biodegradable compounds, such as cyclodextrins (CDs) were employed as a bioavailability enhancer in this work. When the hydroxyl groups of BCD were modified to synthesise HPBCD and RAMEB, the specificity and physicochemical properties were improved, such as the interactions with the pollutant and water solubility. However, in the case of CLP, $K_c$ was higher for BCD than for modified CDs. CLP had a strong lipophilic character and showed a higher affinity for BCD. The hydrophobic cavity size of the CDs used was similar, but the addition of methyl or hydroxypropyl groups may have some effects on the interaction with the hosted organic compound. Therefore, Kc was lower for MBCD and HPBCD, probably due to the presence

of hydroxypropyl groups, which would confer a more hydrophilic character to the cavity, which could result in a decrease in hydrophobic interactions with the insecticide [29]. It should be noted that the $K_c$ of BCD reached a high value, indicating that there existed a strong tendency to form inclusion complexes with CLP; however, $S_e$ was not the highest because BCD has a low solubility in water (16.3 mM) in comparison to HPBCD and RAMEB, which limits the achievement of high $S_e$ values [58].

In this work, RAMEB increased the CLP bioavailable fraction in the soil solution, which implied an improvement in its extent and rate of biodegradation by the endogenous microbiota of the soil [59,60]. Soil CLP extraction experiments showed that RAMEB in the ALC soil could improve the solubility and consequently the bioavailability of the insecticide via complexation [61]. However, the high hydrophobic character of CLP, together with the high OM content of the studied soil (13.9%, Table 1) favoured the formation of very strong links, diminishing the tendency to form inclusion complexes with RAMEB [29,62]. Another possibility could be that RAMEB was also acting as a biostimulant for the soil microbiota activity. This fact has been previously demonstrated for other CDs [25,30,63–65]. Although RAMEB is considered a poorly biodegradable CD [66], Fava et al. [67] observed that aerobic microorganisms isolated from polychlorinated-biphenyls-contaminated soil were able to metabolise RAMEB as only carbon and energy source. As far as we know, there are no scientific studies that show the biodegradation of CLP in soil in the presence of bioaugmentation and CDs. Only Báez et al. [30] carried out CLP biodegradation studies in soils amended with BCD, but without microbial inoculation. In this case, an important enhancement of the microbial activity occurred in the system BCD/CLP, but a more effective degradation of the insecticide was not observed.

The feasibility of the studied biodegradation treatments was checked, carrying out toxicity studies, demonstrating that only a slight decrease in the toxicity at the end of the experiment was observed for the CLP biodegradation in solution.

A similar result was obtained when Echeverri-Jaramillo and Castillo-López [68] studied the toxic effect of CLP and its main metabolite 3,5,6-trichloro-2-pyridinol (TCP) in a solution, and $EC_{50}$ was 0.98 and 3.7 mg $L^{-1}$, respectively, concluding that the final toxicity was due to the presence of the metabolite. This would explain the remaining toxicity that was maintained once we applied our bioremediation treatment in solution. The Microtox® luminimetric test has been used by other authors to study the ecotoxicological effect of CLP in aqueous systems. Mossa et al. [69] observed that the mix of CLP and metabolites was more toxic than CLP. In other study, Jones and Huang [70] evaluated CLP toxicity with and without humic substances compost used as a bioremediation strategy, observing an increase in $EC_{50}$ from 31.57% to 72.01%. However, the complete removal of toxicity was not achieved.

On the contrary, when *B. megaterium* CCLP1 and *B. safensis* CCLP2 were inoculated in ALC-contaminated soil, a drastic decrease in toxicity was observed. This fact would be due to the joint and synergic action of the novel bacterial strains inoculated and the soil endogenous microbiota, which could achieve the degradation of CLP toxic metabolites or the formation of other metabolites less toxic than those formed in solution. It is worth noting that this is the first time that CLP ecotoxicity studies in soils have been published.

## 5. Conclusions

*B. megaterium* CCLP1 and *B. safensis* CCLP2, isolated from two different agricultural soils using enrichment cultures in the presence of CLP, were able to degrade CLP completely in aqueous solution. Three CDs studied (BCD, HPBCD, and RAMEB) were able to increase CLP water solubility significantly, obtaining the best results with RAMEB. For this reason, it was selected as a bioavailability enhancer of CLP in soil, because of its capacity to form an inclusion complex with the insecticide. When the degrading bacterial strains and RAMEB were added together, the best biodegradation results were achieved. Ecotoxicity studies in aqueous solution showed a decline in toxicity when the novel bacterial strains were inoculated, although the complete elimination of toxicity was not reached, indicating that

CLP toxic metabolites were still present. On the contrary, in the case of soil, ecotoxicological studies demonstrated a complete elimination of the toxicity when bioaugmentation and RAMEB treatments were conducted. On this basis, the feasibility of the decontamination strategy proposed for the investigated CLP-contaminated soil could be demonstrated. As a conclusion, bioaugmentation must be considered as a feasible method for CLP remediation in soils, which needs to be adapted to site-specific conditions, and hence, small-scale treatment studies are required before a real site clean-up.

**Supplementary Materials:** The following supporting information can be downloaded at: https://www.mdpi.com/article/10.3390/agronomy12081971/s1, Figure S1: Quantification of the CLP degraded in aqueous solution after 20 d by the bacterial strains isolated from (A) LL soil and (B) R soil, Figure S2: Chlorpyrifos extracted from ALC soil in presence of water, a nutrient solution (NS), and RAMEB.

**Author Contributions:** Conceptualisation, A.L.-M., E.M. and J.V.; methodology, A.L.-M., J.V., F.M. (Francisco Merchán) and F.M. (Fernando Madrid); validation, A.L-M. and J.V.; formal analysis, A.L-M., E.M. and J.V.; investigation, A.L-M., F.M. (Francisco Merchán), F.M. (Fernando Madrid) and J.V.; resources, E.M.; date curation, A.L-M., E.M. and J.V.; writing—original draft preparation, A.L-M.; supervision, E.M., J.V. and F.M. (Francisco Merchán); funding acquisition, E.M. and J.V. All authors have read and agreed to the published version of the manuscript.

**Funding:** This work was supported by the Spanish Ministry of Economy and Competitiveness under the research project CMT2017-82472-C2-1-R (AEI/FEDER, UE).

**Institutional Review Board Statement:** Not applicable.

**Informed Consent Statement:** Not applicable.

**Data Availability Statement:** Not applicable.

**Acknowledgments:** Alba: Lara-Moreno acknowledges the Spanish Ministry of Education, Culture and Sports for her FPU fellowship (FPU15/03740) and University of Seville for her Margarita Salas grant funded by the European Union's Next Generation EU.

**Conflicts of Interest:** The authors declare no conflict of interest.

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
