# Peer review of "Chlorpyrifos Removal in an Artificially Contaminated Soil Using Novel Bacterial Strains and Cyclodextrin. Evaluation of Its Effectiveness by Ecotoxicity Studies"

_agronomy, doi:10.3390/agronomy12081971_

Round 1

Reviewer 1 Report

The article is devoted to the development of a method for bioremediation of soils contaminated with Сhlorpyrifos. This topic is relevant due to the high toxicity and persistence, as well as the widespread use of this insecticide. Of the five soils, two were identified with the highest degradation activity in relation to Сhlorpyrifos, of which two bacterial strains were characterized by the maximum ability to decompose the pesticide. It was shown that the use of these strains together with nutrient solution and Cyclodextrins, which increase the bioavailability of Сhlorpyrifos, contributed to the biodegradation of the insecticide both in solution and in soil. In conclusion, the decrease in soil toxicity as a result of bioremediation was confirmed.

The article has a standard structure: informative introduction, a detailed description of the methods used, a well-structured and understandable presentation of the results, and a conclusion.

The authors conducted a multifaceted scientific study using a large number of different methods. The conclusions drawn were confirmed by the results and compared with the data of other researchers. The work was done at a high professional level, and the article was written clearly and concisely in good English.

The article is recommended for publication without changes.

The only comment available concerns Table 2, where LL soil shows a half-life of 1543 days, while the graph and rate constants indicate that DT50 should be around 200 days.

And in order to discuss the results, the question is:

- A freshly added pesticide was used in the study. Will RAMEB increase the bioavailability of “aged” Chlorpyrifos residues, and what would be the bioremediation efficiency in this case?

There are several comments and questions that do not detract from the importance and high quality of the article.

1. Materials and methods did not specify how CLP-containing samples were purified and concentrated. Or, due to the high application rate, these stages of sample preparation were not required?

2. Table 2 showed a half-life of 1543 days for LL soil, while the graph and rate constants indicated that DT50 should be around 200 days.

3. Perhaps in the table. 2, it would be better for each soil to present an approximation of the degradation dynamics by all three equations.

4. As can be seen from fig. 3, a sharp inflection in the decomposition curve (HS equation) was observed for all variants with the addition of decomposing bacteria. What is it connected with? It seems to me that at the first stage, the bioavailable insecticide degraded quickly, then the rate slowed down due to the fact that the bioavailable CLP was over. It would also be useful to calculate the DT90 for all options to determine how long it will take for 90% of the pesticide to be degraded.

5. Freshly added pesticide was used in the study. Will RAMEB increase the bioavailability of “aged” Сhlorpyrifos residues (as observed in natural conditions), and what would be the bioremediation efficiency in this case?

6. Why was such a high rate of CLP used in the experiment? Does this mean that this bioremediation method is supposed to be used only for extremely polluted soils (accidental spillage, etc.), or is it also planned to be used for soils where CLP was applied at the recommended rates?

7. How likely do you think that high rates of the insecticide will be adsorbed weaker than low doses due to the limited number of sorption sites? And therefore, would bioremediation of low amounts of CLP be less effective?  

8. In the study, samples were incubated at 30°C. How will the pesticide be degraded by bacteria at lower temperatures? The same applies to soil with low moisture, since in laboratory conditions it was optimal for bacterial growth.

Author Response

Agronomy

14th August 2022

Dear Editor,

I enclose the revised version of the manuscript entitled: “Chlorpyrifos removal in an artificially contaminated soil using novel bacterial strains and cyclodextrin.  Evaluation of its effectiveness by ecotoxicity studies.”, by Alba Lara-Moreno, Esmeralda Morillo, Francisco Merchán, Fernando Madrid, Jaime Villaverde. ID: Agronomy-1874626.

The revised manuscript with the changes marked is submitted. All the corrections suggested have been included. It has been revised according to the reviewers suggestions, using the “Track Changes” function how was pointed out in the previous email, in order to be easily viewed by the reviewers.

We are very grateful for the comments made by the reviewers, which have improved the manuscript from a scientific and stylistic point of view.

We now hope that you find it suitable for publication in Agronomy Journal for the Special Issue "Impact of Agrochemicals on Soil“.

Sincerely,

Alba Lara

Dra. Alba Lara Moreno.

Instituto de Recursos Naturales y Agrobiología (CSIC),

Apdo 1052, 41080-Sevilla, Spain

E-mail: lara@irnas.csic.es

Response to Reviewers

Reviewer #1:

The article is devoted to the development of a method for bioremediation of soils contaminated with Сhlorpyrifos. This topic is relevant due to the high toxicity and persistence, as well as the widespread use of this insecticide. Of the five soils, two were identified with the highest degradation activity in relation to Сhlorpyrifos, of which two bacterial strains were characterized by the maximum ability to decompose the pesticide. It was shown that the use of these strains together with nutrient solution and Cyclodextrins, which increase the bioavailability of Сhlorpyrifos, contributed to the biodegradation of the insecticide both in solution and in soil. In conclusion, the decrease in soil toxicity as a result of bioremediation was confirmed.

The article has a standard structure: informative introduction, a detailed description of the methods used, a well-structured and understandable presentation of the results, and a conclusion.

The authors conducted a multifaceted scientific study using a large number of different methods. The conclusions drawn were confirmed by the results and compared with the data of other researchers. The work was done at a high professional level, and the article was written clearly and concisely in good English.

The article is recommended for publication without changes.

The only comment available concerns Table 2, where LL soil shows a half-life of 1543 days, while the graph and rate constants indicate that DT50 should be around 200 days.

Response:

The kinetic model has been reviewed and the result is correct. After 20 days of degradation a stationary phase begins and for that reason it takes more than 1500 days to reach 50% of the degradation.

And in order to discuss the results, the question is:

- A freshly added pesticide was used in the study. Will RAMEB increase the bioavailability of “aged” Chlorpyrifos residues, and what would be the bioremediation efficiency in this case?

Response:

This question is interesting, for this reason studies with RAMEB in chlorpyrifos contaminated soils aged at different times (30, 60 and 100 days) are planned to be performed.

There are several comments and questions that do not detract from the importance and high quality of the article.

  1. Materials and methods did not specify how CLP-containing samples were purified and concentrated. Or, due to the high application rate, these stages of sample preparation were not required?

Response:

In this work it was not necessary to purify or concentrate the samples because a concentration of 10 mg L-1 was initially added. Our HPLC has enough sensitivity to quantify the worked concentrations.

  1. Table 2 showed a half-life of 1543 days for LL soil, while the graph and rate constants indicated that DT50 should be around 200 days.

Response:

The kinetic model has been reviewed and the result is correct. After 20 days of degradation a stationary phase begins and for that reason it takes more than 1500 days to reach 50% of the degradation, although visually it may seem that 50% degradation is reached sooner.

  1. Perhaps in the table. 2, it would be better for each soil to present an approximation of the degradation dynamics by all three equations.

Response:

Biodegradation curves were fitted to three first-order kinetic models. However, the most appropriate kinetic model (smaller χ 2) was selected. The results of all models have been shown in the following table. In some cases, it has been impossible to fit the biodegradation curve to the kinetic model (indicated with error). Only the best model has been included in the manuscript (red colour).

Soil

Kinetic model

K1

(d-1)

K2

(d-1)

tb

(d)

α

(d-1)

β

(d-1)

DT50

(d)*

% Extension of mineralization

Lag phase (d)**

χ 2***

LL

SFO

4.5x10-3

-

-

-

-

151,2

52.59

5

12.5

LL

FOMC

-

-

-

ERROR

ERROR

-

-

-

ERROR

LL

HS

4.5x10-2

1.4x10-3

15.7

-

-

1573

47.3

5

1.29

PLD

SFO

4.7x10-3

-

-

-

-

144.4

35.2

11

2.7

PLD

FOMC

-

-

-

0.3

0.4

485

33.2

11

0.5

PLD

HS

5.7x10-4

4.4x10-3

7.2

-

-

163

33.8

11

1.8

R

SFO

9.2x10-3

-

-

-

-

74.9

70.4

4

19.8

R

FOMC

-

-

-

0.2

1.2

20

64.1

4

9

R

HS

7.2x10-2

2.0x10-3

11.7

-

-

9.6

61.4

4

1.63

ALC

SFO

1.6x10-4

-

-

-

-

4203

0.7

-

0.003

ALC

FOMC

-

-

-

ERROR

ERROR

-

-

-

ERROR

ALC

HS

ERROR

ERROR

-

-

-

-

-

-

ERROR

CR

SFO

4.7x10-4

-

-

-

-

1474

4.5

-

0.02

CR

FOMC

-

-

-

ERROR

ERROR

-

-

-

ERROR

CR

HS

ERROR

ERROR

-

-

-

-

-

-

ERROR

  1. As can be seen from fig. 3, a sharp inflection in the decomposition curve (HS equation) was observed for all variants with the addition of decomposing bacteria. What is it connected with? It seems to me that at the first stage, the bioavailable insecticide degraded quickly, then the rate slowed down due to the fact that the bioavailable CLP was over. It would also be useful to calculate the DT90 for all options to determine how long it will take for 90% of the pesticide to be degraded.

Response:

Effectively, in the case of the HS model k1 and k2 are the rate constants of degradation for the fast and the slow fraction, respectively, and tb is the time at which rate constant changes. K1 shows a quick degradation and k2 slower degradation possibly due to a severe bioavailability decline of the insecticide.

DT90 values were added in Table 4.

Information added to the manuscript (lines 668-672).

  1. Freshly added pesticide was used in the study. Will RAMEB increase the bioavailability of “aged” Сhlorpyrifos residues (as observed in natural conditions), and what would be the bioremediation efficiency in this case?

Response:

This question is interesting, for this reason studies with RAMEB in chlorpyrifos contaminated soils aged at different times (30, 60 and 100 days) are planned to be performed. In the present study we concluded that RAMEB is able to increase the water solubility 311 (±1.1) times. For this reason, it is expected to improve the bioremediation efficiency in presence of RAMEB, even in aged residues, although this will depend on the soil properties, for instance, the soil organic matter content.

  1. Why was such a high rate of CLP used in the experiment? Does this mean that this bioremediation method is supposed to be used only for extremely polluted soils (accidental spillage, etc.), or is it also planned to be used for soils where CLP was applied at the recommended rates?

Response:

50 mg Kg-1 has been added to simulate pollution events, such as the disposal or accidental release of high pesticide concentrations (Villaverde et al., 2018; Rodríguez-Cruz et al., 2019). In such situations, relatively small plots of soils could be treated by bioaugmentation. Information was included in lines 329-331.

Rodríguez-Cruz, M. S.; Pose-Juan, E.; Marín-Benito, J. M.; Igual, J. M.; Sánchez-Martín, M. J. Pethoxamid dissipation and microbial activity and structure in an agricultural soil: Effect of herbicide rate and organic residues. App. Soil Ecol. 2019, 140, 135–143. https://doi.org/10.1016/j.apsoil.2019.04.011.

Villaverde, J.; Rubio-Bellido, M.; Lara-Moreno, A.; Merchan, F.; Morillo, E. Combined use of microbial consortia isolated from different agricultural soils and cyclodextrin as a bioremediation technique for herbicide contaminated soils. Chemosphere, 2018, 193, 118–125. https://doi.org/10.1016/j.chemosphere.2017.10.172.

  1. How likely do you think that high rates of the insecticide will be adsorbed weaker than low doses due to the limited number of sorption sites? And therefore, would bioremediation of low amounts of CLP be less effective?

Response:

The main purpose of this study is the bioremediation of contaminated soil in an occasional or accidental way. But indeed, with reduced concentrations, the greater the number of active adsorption points with strong affinity will make more difficult CLP desorption, and, therefore its availability to be biodegraded.

  1. In the study, samples were incubated at 30°C. How will the pesticide be degraded by bacteria at lower temperatures? The same applies to soil with low moisture, since in laboratory conditions it was optimal for bacterial growth.

Response:

The reviewer is right and bioremediation techniques will be affected by soil properties and environmental conditions, especially temperature and moisture, as indicated in the conclusions of the manuscript.

Reviewer 2 Report

The subject of the manuscript is consistent with the scope of the Journal. The topic of research is interesting. The paper is well written and technically sound. It has a thorough testing program and it adds value. Some revisions are needed for improving the manuscript.

 1.     Soil grain size composition was assessed according to USDA or IUSS? Enter the particle sizes for sand, silty and clay.

2.     Give more details on the determination of organic matter.

3.     Divide "3. Results and discussion" into two separate sections: "3. Results" and "4Discussion".

4.     Figure S1 should be listed in Supplementary Materials.

5.     All abbreviations used in the manuscript should be explained in the text when they are first used and under the tables.

6.     Please, be sure that all the references cited in the manuscript are also included in the reference list and vice versa with matching spellings and dates.

Author Response

Agronomy

14th August 2022

Dear Editor,

I enclose the revised version of the manuscript entitled: “Chlorpyrifos removal in an artificially contaminated soil using novel bacterial strains and cyclodextrin.  Evaluation of its effectiveness by ecotoxicity studies.”, by Alba Lara-Moreno, Esmeralda Morillo, Francisco Merchán, Fernando Madrid, Jaime Villaverde. ID: Agronomy-1874626.

The revised manuscript with the changes marked is submitted. All the corrections suggested have been included. It has been revised according to the reviewers suggestions, using the “Track Changes” function how was pointed out in the previous email, in order to be easily viewed by the reviewers.

We are very grateful for the comments made by the reviewers, which have improved the manuscript from a scientific and stylistic point of view.

We now hope that you find it suitable for publication in Agronomy Journal for the Special Issue "Impact of Agrochemicals on Soil“.

Sincerely,

Alba Lara

Dra. Alba Lara Moreno.

Instituto de Recursos Naturales y Agrobiología (CSIC),

Apdo 1052, 41080-Sevilla, Spain

E-mail: lara@irnas.csic.es

Response to Reviewers

Reviewer #2:

The subject of the manuscript is consistent with the scope of the Journal. The topic of research is interesting. The paper is well written and technically sound. It has a thorough testing program and it adds value. Some revisions are needed for improving the manuscript.

  1. Soil grain size composition was assessed according to USDA or IUSS? Enter the particle sizes for sand, silty and clay.

Response:

Taxonomic classifications have been added also in Table 1 are based on USDA Soil Taxonomy and the information has been obtained from Soil Maps, 2005. National Geographic Institute. Nature Database (Spanish Ministry of Environment). https://www.ign.es/web/catalogo-cartoteca/resources/html/030769.html

  1. Give more details on the determination of organic matter.

Response:

Information about organic matter determination has been included (Line 132-133).

  1. Divide "3. Results and discussion" into two separate sections: "3. Results" and "4Discussion".

Response:

Results and discussion section has been separated.

  1. Figure S1 should be listed in Supplementary Materials.

Response:

Figure S1 is listed in supplementary materials (Line 738).

  1. All abbreviations used in the manuscript should be explained in the text when they are first used and under the tables.

Response:

All abbreviations have been reviewed (Line 43, 98, 138, 204, 241, 318, 342, 404, 489).

  1. Please, be sure that all the references cited in the manuscript are also included in the reference list and vice versa with matching spellings and dates.

Response:

The references cited have been reviewed.
